# Tea: Transfer of Mycotoxins from the Spiked Matrix into an Infusion

**DOI:** 10.3390/toxins13060404

**Published:** 2021-06-07

**Authors:** Mariya Kiseleva, Zakhar Chalyy, Irina Sedova

**Affiliations:** Federal Research Centre of Nutrition and Biotechnology, Ust’inskiy pr., 2/14, 109240 Moscow, Russia; brew@ion.ru (Z.C.); isedova@ion.ru (I.S.)

**Keywords:** mycotoxins, *Camellia sinensis* and herbal tea, infusions, transfer rate, transfer kinetics, water TDS, infusion pH, HPLC-MS/MS

## Abstract

Recent surveys report the occurrence of *Aspergillus* and *Penicillium* metabolites (aflatoxins (AFLs), ochratoxin A (OTA), cyclopiazonic and mycophenolic acids (MPA), sterigmatocystin (STC), citrinin), *Fusarium* (trichothecenes, zearalenone (ZEA), fumonisins (FBs), enniatins (ENNs)) and *Alternaria* (alternariol (AOH), its methyl ether (AME), tentoxin (TE), and tenuazonic acid (TNZ)) toxins in dry *Camellia sinensis* and herbal tea samples. Since tea is consumed in the form of infusion, correct risk assessment needs evaluation of mycotoxins’ transfer rates. We have studied the transfer of AFLs, OTA, STC, deoxynivalenol (DON), ZEA, FBs, T-2, and HT-2 toxins, AOH, AME, TE, ENN A and B, beauvericin (BEA), and MPA from the spiked green tea matrix into an infusion under variation of preparation time and water characteristics (total dissolved solids (TDS) and pH). Analytes were detected by HPLC-MS/MS. The main factors affecting transfer rate proved to be mycotoxins’ polarity, pH of the resulting infusion (for OTA, FB2, and MPA) and matrix-infusion contact period. The concentration of mycotoxins increased by 20–50% within the first ten minutes of infusing, after that kinetic curve changed slowly. The concentration of DON and FB2 increased by about 10%, for ZEA, MPA, and STC it stayed constant, while for T-2, TE, AOH, and AFLs G1 and G2 it went down. Maximum transfer correlated well with analytes polarity. Maximum transfer of ENNs, BEA, STC, ZEA, and AOH into infusion was below 25%; AFLs—25–45%; DON, TE, and T-2 toxins 60–90%, FB1—80–100%. The concentration of OTA, MPA, and FB2 in the infusion depended on its pH. At pH about four, 20%, 40%, and 60% of these toxins transferred into an infusion, at pH about seven, their concentrations doubled. Water TDS did not affect transfer significantly.

## 1. Introduction

Tea is a worldwide daily basic food product. FAO Intergovernmental Group on Tea reports *C. sinensis* tea consumption is growing. People increase their awareness related health impacts of carbonated drinks, choosing tea as an alternative [1]. *C. sinensis* and herbal teas are traditionally considered “helpful or at least harmless” for health promotion. Turkey, Libya, Morocco, Ireland, and the UK demonstrate the highest per capita consumption of *C. sinensis* tea, China, the Russian Federation, Japan, and India are in the list of the most important consuming countries [2]. Mean daily black tea intake within all population of Russia was 446 mL per capita, according to the 1995–2003 study [3]. EFSA Comprehensive European Food Consumption Database reports *C. sinensis* tea average chronic daily consumption by adults equal to 124 g (min: 5 g, max: 406 g, median: 100 g) per capita; “herbal and other non-tea infusions” 61 g (min: 4 g, max: 217 g, median: 38 g) per capita [4].

Recent surveys revealed the problem of herbal and *C. sinensis* tea contamination with micromycetes and their metabolites. Selected literature data on the occurrence of mycotoxins are summarized in Table 1.

Mannani et al. reported that about 60% of 129 green tea samples purchased in Morocco were AFLs positive, the concentration of mycotoxins in contaminated samples reached dozens of μg/kg [5]. Meanwhile, El Jai et al. screened 111 green tea samples of the same origin for AFLs, ENNs, BEA, OTA, ZEA, AOH, and TE and detected AOH, ZEA, AFL G1 and G2, and ENNB [6]. A total of 40% and 35% samples were contaminated with AOH and ZEA, and maximum concentrations reached 5.9 μg/kg and 45.8 μg/kg, correspondingly. Occurrence and contamination levels of AFLs were much lower in comparison with the results of the previous study. A total of 45 black and 15 green tea samples from Iran were tested for OTA and AFLs [7], respectively. A total of 91% and 73% of samples were OTA positive; mean concentration and range were as follows: 6.1/5.1–30.9 μg/kg and 2.8/0.5–20.4 μg/kg. A total of 40% and 13% of samples were contaminated with AFLs (1.5/0.7–4.2 μg/kg and 1.9/1.0–2.8), respectively. Low contamination regulated in foods mycotoxins was reported for post-fermented Chinese teas [8,9]. Only two of 158 samples tested by Cui et al. were AFLs positive; one sample contained 2.1 μg/kg AFL B1, another—3.9 μg/kg total AFLs [8]. Ye et al. inspected 108 samples of Chinese dark post-fermented tea for ZEA, AFLs, OTA, DON, FBs, and T-2 toxin (T-2): only OTA and ZEA were revealed over the limit of detection (LOD). A total of 4.6% samples were ZEA-positive (5.3/<LOD—182.3 μg/kg); OTA was quantified in 1.9% sample s (0.7/<LOD—36.4 μg/kg) [9]. Reinholds et al. applied 2D-HPLC-MS/MS for the detection of 70 fungal metabolites in 20 Pu-erh teas available in Latvia [10]. DON and its derivatives occurred in 100% samples (up to 9 mg/kg DON), AFL B1, B2, and G1 were detected in 30–40% of samples. AFL B1 concentration was up to 9.2 μg/kg. A total of 40% of samples were STC-positive, its concentration reached 93.4 μg/kg. ZEA was detected in 35% samples, maximum level—56 μg/kg. OTA contaminated a quarter of studied samples (≤4.1 μg/kg). The occurrence of *Alternaria* mycotoxins and emerging *Fusarium* toxins ENNs was 40–55%. Plenty of *C. sinensis* black and green tea samples were also tested in this study. Detection of T-2 and HT-2 toxins in two green tea samples should be noted. DON, T-2, and HT-2 mycotoxins are contaminants common for cereals. It is rather unexpected to find high concentrations of trichothecenes in tea samples. However, their occurrence in herbs has been reported [16,17] (Table 1). Risk assessment carried out by Reinholds et al. showed, that high DON and its derivatives content in tea could provide potential exposure up to 79% of tolerable daily intake [10]. Multi-mycotoxin screening of 54 samples of loose and bagged *C. sinensis* tea available in the Russian Federation revealed low contamination: none of the studied mycotoxins was detected above limit of quantification (LOQ); traces of MPA (excluding several samples with concentration up to 200 µg/kg) and BEA were detected in 26% and 58% of studied samples, correspondingly [11].

Compared to *C. sinensis*, herbal teas are usually contaminated with mycotoxins at higher levels. In the latter survey ZEA, STC, MPA, *Alternaria* toxins, and ENN B were detected in 10 out of 11 examined multi-component herbal tea samples ranging from several μg/kg to several mg/kg. The occurrence of mycotoxins in Chinese herbal medicines was well-reviewed in [12]. Roots and rhizomes were the most susceptible to contamination. Fruits and seeds followed them. According to the summarized data, 23.4% of about three thousand samples were positive for AFL B1. Its concentration varied in the range from 0.02 to 1268.8 μg/kg. A total of 26% of 303 tested samples contained OTA in the range from 0.01 to 158.7 μg/kg. ZEA was detected in 41% of 105 samples (0.2 to 931.07 μg/kg). The authors summarize scarce data on FBs, DON, and T-2 occurrence also. Alongside AFLs and OTA, Chen et al. reported up to 53 μg/kg CIT in Chinese traditional medicinal herbs [13]. TNZ was a major *Alternaria* secondary metabolite in 37 samples of herbs marketed in Lebanon [14]. Its concentration reached 2.5 mg/kg. Duarte et al. detected AFLs and ZEA in teas and medicinal plants destined to prepare infusions in Portugal [15]. Approximately 65% of the 37 analyzed samples were positive for AFLs (2.8–28.2 μg/kg); 62% were contaminated with a ZEA (1.8–19.0 μg/kg). Altyn and Twaruzek reviewed mycotoxin occurrence in the principal herbal components (liquorice, chamomile, mint, ginseng, milk thistle, and ginger) and existing regulations [18]. They pointed out that AFLs and OTA are the most frequently detected mycotoxins in herbal products marketed in the EU: these toxins have repeatedly occurred at concentrations exceeding regulatory levels.

Thus, data on mycotoxins in *C. sinensis* and herbs are diverse and sometimes controversial. Regulations concern mainly AFLs and OTA: maximum level (ML) was set in Russia for AFL B1 in tea (5 μg/kg), in Argentina for AFL B1 and AFLs (5 and 20 μg/kg correspondingly), in Japan, China, Sri Lanka, and India in the category “all foods” [19]. AFLs and OTA are regulated by the EU in ginger, liquorice root and extracts [18]. MLs for AFLs in herbal drugs were set by the European Pharmacopeia, while herbal teas and food supplements are not subjected to control (except mentioned above-selected species). Risk assessments for mycotoxins in *C. sinensis* and herbal tea are traditionally carried out basing on their concentration in dry tea. However, the input of tea into the intake is due to infusion only. It is reasonable to determine the transfer rate of mycotoxins into an infusion. There are only a few data concerning this issue available in the literature. Three studies considered evaluation of OTA transfer from contaminated dry tea and coffee into beverage [20,21,22]. It varied from 4 to 66% depending on the matrix and preparation procedure. Monbaliu et al. looked for FB1 in the infusion of naturally contaminated *C. sinensis* tea sample. According to the authors, they failed to detect it there, possibly due to lack of sensitivity [23]. Reinholds et al. evaluated DON and ZEA transfer into infusions of medicinal herbal teas at the level of 32–100% and 100% correspondingly [24]. We have studied the transfer of OTA, STC, MPA, ZEA, ENN A and B, BEA, and three *Alternaria* toxins (TE, AOH, and AME) from naturally contaminated herbal tea into infusion [25]. Mycotoxins’ polarity and infusion pH (for analytes possessing carboxylic groups) appeared to determine the transfer rate. The maximum transfer was revealed for MPA (96%), minimum—for ENNs and BEA (<7%). Abd El-Aty et al. summarized literature data concerning the fate of various contaminants, including pesticides, polycyclic aromatic hydrocarbons, mycotoxins, and heavy metals in tea infusions [26]. The authors referred to the only one available at the time study by Monbaliu et al. [23], who did not detect notable FB1 transfer into the infusion of naturally contaminated tea. Meanwhile, discussion of pesticides’ transfer revealed infusing period and contaminants hydrophobicity as factors determining transfer rate.

The present study aimed to determine the transfer of 20 mycotoxins from spiked *C. sinensis* green tea matrix into infusion prepared using distilled, deep-well, and mineral water with pH adjusted to the value in the range from 3 to 9. Mycotoxins transfer kinetics was monitored for 30 min. The outcomes could provide basis for accurate risk assessment of teas and, probably, evidence-based recommendations for infusions safety insurance.

## 2. Results and Discussion

Numerous parameters define the procedure of a tea infusion preparation: type of tea (green, black, Pu-erh, or any other *C. sinensis* tea; herbal tea; bagged or loose tea), traditions, individual preferences, characteristics of water and teapot material and even atmospheric pressure. There hardly can exist any common methodology. Moreover, the International Standard for preparation of tea liquor for use in sensory tests (ISO 3103-2019 [27]) defines the proportion of dry tea and water (2 g for 100 mL), amount of time for infusing (6 min), pot material and other details concerning sensory tests. It is noted that tests results are usually affected by the hardness of the water used for preparation. Distilled or deionized water may be used in the case of comparative studies. The temperature of water used in tests is described as “freshly boiled”.

We have studied mycotoxins’ transfer as a function of the infusion-matrix contact period, water TDS, and infusion pH. Loose green tea has been chosen as a matrix. First, because it is a popular kind of tea and we have plenty of “clean” sample. Second, green tea extracts yield lower MS/MS signal suppression than black and herbal teas according to our experience. The following parameters stayed constant within the present study: the proportion of dry tea and water was set according to ISO recommendations (2 g for 100 mL). The infusions were prepared in glass beakers by pouring water heated to 99.9 °C.

### 2.1. The Effect of Infusing Period on Mycotoxins Transfer

As has been mentioned above, there are not any strict guidelines concerning the duration of the infusion procedure. Period of contact between tea leaves and infusion usually varies from several minutes in the case of bagged tea to several dozens of minutes needed for herbal tea preparation. Recovery of mycotoxins from the tea matrix is time dependent. Malir et al. demonstrated it for OTA: 15 min contact yielded 41.5 ± 7% transfer, while within 3 min 34.8 ± 1.3% of toxin transferred into infusion [20,21].

In the present study, we varied the contact period from zero (to be more precise, a dozen seconds: test sample was taken immediately after pouring hot water into the beaker with spiked tea leaves and mixing) to 30 min (Figure 1, Appendix A). The concentration of most of the mycotoxins increased within the first ten minutes by 20–50% and then stayed constant or changed by less than 10%. DON, ALT, FB1, and FB2 transfer into infusion elevated to over 90% and was growing within the whole tested period. The maximum concentration of hydrophobic toxins, such as STC, ZEA, ENNs, and BEA, in the infusion was detected at 5–10 min; it corresponded to 23%, 28%, 10–14%, and 6%, respectively. Then it decreased due to water cooling or reabsorption. A similar decrease in 5–8% was observed for AFL G1, T-2, TE, and AOH. The concentration of AFL G2 in the infusion reduced from 73 to 57% of spiking level since the 10th minute. This is probably due to toxin’s degradation in water solutions [28].

This model experiment showed that quick discard of the first infusion portion followed by subsequent re-infusion can result in 20–85% elimination of mycotoxins; possible reduction of 40% and 64% of initial level of the most potent hazards—AFL B1 and OTA, respectively—is possible (Appendix A). At the same time, antioxidant capacity of re-infused tea leaves has been proved to remain still high [29].

Thirty minutes infusing period was chosen for further studies because the transfer of the most of studied mycotoxins (14 of 19) changed by less than 10% since the 20th minute. In addition, prolonged infusing is usually recommended for herbal tea preparation, and they are the most at risk for mycotoxins contamination [11,16,18,24].

### 2.2. Mycotoxins Transfer from the Bare Spike

Transfer from the bare spike in the absence of matrix was studied to evaluate mycotoxins’ losses due to degradation or adsorption on glass surface during infusion preparation procedure (Figure 2, Appendix A). Distilled water pH was adjusted to three, seven, and nine by adding citric acid and sodium carbonate solutions.

Degradation in the infusion can be thermal or chemical. Mycotoxins are generally considered heat-stable molecules. Prolonged exposure to high temperatures common to baking, frying, roasting, and extrusion is a well-studied food and feed detoxification approach [30,31]. Partial degradation can occur under conventional cooking temperatures. Traditional rice cooking was reported to result in 30–50% AFL B1 degradation; buffered solutions of FBs and DON (excluding solutions with pH about 10) were indifferent to heating [32]. Similar results were obtained for OTA: its heating in deionized water and buffered solutions with pH 4 and 7 at 100 °C for 60 min did not cause notable degradation, while at pH 10 the concentration declined down to 25% of the initial level [33]. The fate of mycotoxins during pasta preparation was summarized in [34], it is reported that reduction of DON and OTA was due to transfer to cooking water; degradation products were detected for ENNs. There are only few data concerning thermal stability of *Alternaria* toxins: wet baking at 160 °C for 60 min did not change concentration of AOH, AME, and altenuene (ALT) according to [35], while in [36] a 35% decrease of AOH was detected. To the best of our knowledge there is no data concerning degradation of mycotoxins under tea infusion preparation conditions. Basing on the above short summary of literature data, we can assume that under acidic and neutral conditions most of the studied mycotoxins would hardly undergo substantial thermal degradation. Chemical degradation in water solutions at pH about 9 has been earlier reported in literature for T-2 and HT-2 toxins, G-group AFLs, and AOH [28]. Thus, mycotoxins recovery from bare spike into acidic and neutral infusions is expected to be governed first of all by their solubility. Possible losses due to affinity to glass surfaces, and chemical and thermal degradation were also expected to be revealed.

Recovery from the bare spike of the most of neutral mycotoxins (AFL B1, AFL B2, STC, DON, T-2, and HT-2) and several acidic ones with pKa > 7 (ZEA, ALT, TE, and AOH) varied in the range from 60 to 95%. It was almost pH-independent within the margins of uncertainty. AOH was an exception: an increase of pH up to 9 resulted in recovery growth up to 82%. We could not determine AME transfer rate because its concentration in the infusion was below LOQ; hence, it was <12.5%. A decrease of AFLs G series concentration in solution was observed with pH growth from 3 to 7 and 9. This is consistent with literature data on AFLs stability: (1) AFLs G series were found to be less stable in water-organics solutions in comparison with B series; (2) acidification contributes positively to stability [28].

Transfer of acidic mycotoxin MPA (pKa 4.2) increased with pH from 67 to 77%. The acidity of water used for infusions preparation had an unexpectedly low effect on its transfer rate. Meanwhile, for amphoteric OTA and FBs it was well pronounced: at pH 7 and 9 the concentration of these mycotoxins in the infusion was 22% and 50–64% higher than at pH about 3.

An increase of recovery with pH was noted for ENNs and BEA. At pH 3 it was about 44% for ENN B and 20% for ENN A and BEA, at pH 9 it doubled. This may be due to the ionophoric properties of these cyclic hexadepsipeptides. As far as pH 9 of water used for the preparation of the infusions was adjusted with sodium bicarbonate, the concentration of sodium cations increased, enabling the formation of [MNa]^+^ complexes.

Thus, recovery of the most studied mycotoxins at pH 7 was over 60%. It exceeded 80% for two of the most polar studied mycotoxins—DON and ALT pH affected the transfer of OTA, FBs, ENNs, and BEA. A decrease of AFL G1 and AFL G2 concentrations in the infusions with pH 7 and 9 can be due to their degradation.

### 2.3. Mycotoxins Transfer into an Infusion of Spiked Tea

We have examined the transfer of mycotoxins from spiked dry green tea matrix into infusion using three types of water: distilled (DW), deep-well (D-WW), and mineral (MW) with TDS values equal to 4.4, 155, and 238 mg/L, correspondingly. Before infusion preparation, water pH was adjusted to 3, 4, 5, 6, 7, 8, and 9. Acidic and basic compounds from the tea matrix (polyphenols, amino acids, etc.) changed the resulting pH of infusion. It was most noticeable for DW infusions. The resulting pH range for DW infusions was 5.1–7.4, for D-WW infusions—4.0–7.1, for NMW infusions—3.8–7.0 (Appendix A). It might be correct to discuss the transfer rate of mycotoxins under variation of water salinity and infusion pH (not water used for its preparation). AME, ENN A, and BEA have not been quantified in the infusions; their maximum transfer rate was evaluated using LOQ. The whole pool of results is available in Appendix A.

#### 2.3.1. The Effect of Water TDS

Water salinity up to 240 mg/L TDS did not affect the transfer rate of all studied mycotoxins within measurement uncertainty. DON and ENN B were the only exclusions (Appendix A). The average transfer rate for DON when preparing infusion using DW and D-WW proved to be 83 ± 5%. A slight decrease was observed for NMW—70 ± 5%. The average transfer of ENN B into DW infusions was 12 ± 5%, increase of water TDS resulted in transfer decline down to 3% due to salting out of hydrophobic molecules. Supposedly, variation of transfer with water TDS could occur for ENN A and BEA, possessing similar structures. However, the latter could not be quantified in the infusion due to their low transfer rate.

#### 2.3.2. The Effect of Infusion pH

The transfer of the most of studied mycotoxins into an infusion of the spiked matrix was pH indifferent. The only exceptions were AFL G1 and AFL G2, T-2 toxin, MPA, OTA, and FB 2 (Figure 3, Appendix A). The concentration of G-type AFLs and T-2 decreased at pH ≥ 7 by 20–30% compared to infusions with pH 3.8–6.7. A similar decrease has been noted and discussed above for AFL G1 and G2 in the bare spike. Degradation of T-2 by cleavage of ester side chains at pH above 7 [28] could be the reason; however, it did not occur in bare spike infusion at pH 9.

A higher transfer rate was observed for acidic and amphoteric molecules with pKa 3.2–4.2 (Table 2). OTA, MPA, and FB2 transfer rates changed from about 40% to 70–90% due to their carboxylic groups dissociation with pH growth from 3.8 to 7.4. FB1, as well as DON, are the most hydrophilic molecules among studied mycotoxins. Their high polarity provided high transfer rates; even dissociation of FB1 carboxylic groups could not enhance it. The concentration of FB1 in spiked matrix infusions varied between 80 and 100% in the whole studied pH range, while for the bare spike, the decrease of FBs transfer at pH 3 was dramatic (down to 37% and 26% for FB1 and FB2, correspondingly). Probably the reason was that the pH of tea infusions differed from that of the bare spike. The lowest pH was observed for green tea infused with NMW; it was 3.84.

Transfer rate of DON, FB1, T-2 (at pH below 7), and HT-2 toxins, ZEA, AFLs (AFL G1 and G2 at pH below 7), STC, ALT, TE, AOH, and ENN B did not depend on infusion pH. It is not surprising for neutral molecules, as these compounds do not possess functional groups able to get ionized in the studied pH range. Mycotoxins, possessing acidic groups with pKa > 7 (ZEA, ALT, AOH, and AME), could not change their charge much, as maximum infusion pH was 7.3. Variation of the transfer rate of studied mycotoxins, excluding OTA, MPA, and FB2, with infusion pH and water TDS was almost within detection repeatability. Thus we used combined data of the whole studied pH and TDS range to calculate their average transfer rate. Several outliers were discarded (they are marked as empty symbols in Figure 3 for FB1 and Appendix A). We have obtained 21 points for each mycotoxin (for AFLs G group and T-2—at least 15) in triplicate, except outliers and cases when analyte could not be quantified. These data were used for the average transfer rate calculation (Table 2).

Concentrations of ENN B in D-WW and NMW infusions, AME, ENN A, and BEA in all studied infusions were evaluated as <LOQ, transfer rates were calculated accordingly. The average transfer into infusion proved to correlate well with mycotoxins polarity (Figure 4). We used XLogP3-AA values from PubChem Database [36] because they were calculated uniformly and are available for all studied mycotoxins.

Transfer proved to be in inverse proportion to hydrophobicity, R^2^ of exponential dependence was 0.86. Studied mycotoxins can be grouped in four clusters. The first one includes polar mycotoxins easily transferring into water infusions (DON, T-2, and HT-2 toxins, ALT, TE, and FB1). FB2 joins this group for infusions with pH over 5. The transfer rate for these mycotoxins was over 70%. Supposedly, high recovery of such regulated mycotoxins as DON, FBs, T-2, and HT-2 toxins into water infusions can be used for their extraction from various food and feed matrixes. In the absence of organic solvents, resulting extracts would contain lass hindering analyte quantification matrix components compared to traditional approaches. The second cluster consists of AFLs, whose transfer is in the range from 30 to 50%, the third one—ZEA, AOH, AME, and STC, characterized by 10–20% transfer rate. Less than 5% of ENNs (excluding ENN B in DW infusions) and BEA can be found in the infusion.

Transfer of OTA, FB2, and MPA is pH-dependent and varies from 38 to 103% in the pH range from 3.8 to 7.4. Thus, a slice of lemon can probably halve risks associated with these mycotoxins in tea.

## 3. Conclusions

The transfer rate of mycotoxins from spiked green tea matrix into infusion proved to be determined by infusion period, mycotoxins’ polarity, and infusion pH. Salinity effect could be considered negligible for water TDS up to 240 mg/L. Maximum increase of mycotoxins concentration in the infusion occurred within the first ten minutes of infusion preparation. Infusion pH determined transfer of OTA, MPA, and FB2, for all other mycotoxins this factor was irrelevant. Based on experimental data, average transfer rates were calculated for these mycotoxins. Correlation between average transfer and mycotoxins polarity was characterized by R^2^ = 0.86.

## 4. Materials and Methods

### 4.1. Materials and Chemicals

Water was purified using a Milli-Q system (Millipore, Merck Russia, Moscow, Russia). Formic acid (pure, 98+%, Acros organics, Germany), ammonium formate (for LC-MS, Sigma Aldrich, Moscow, Russia), HPLC-grade methanol (LiChrosolv, Merck, Germany), and acetonitrile (Panreac, AppliChem, Germany) were used for the preparation of the standard solutions of mycotoxins and mobile phases. Acetonitrile (pure, Criochrom, St. Petersburgh, Russia) was used for sample extraction. The pH of water for tea infusing was adjusted by citric acid monohydrate (≥99.8%, Lenreactiv, St-Petersburg, Russia), sodium hydroxide (>98%, Reakhim, Moscow, Russia), sodium bicarbonate (99.5–100.5%, Sigma Aldrich, Moscow, Russia).

Three types of water were used to prepare infusions: distilled water (DW) (average pH 5.4 (*n* = 6, CV = 2.1%); total dissolved solids, average TDS = 4.4 mg/L (*n* = 5, CV = 12%)), bottled deep-well water (D-WW) (average pH 7.9 (n = 6, CV = 6,4%); average TDS = 155 mg/L (*n* = 5, CV = 0.8%)) and bottled still natural mineral water (NMW) (average pH 7.6 (*n* = 5, CV = 2.2%); average TDS = 238 mg/L (*n* = 5, CV = 0.4%)). TDS were measured by portable conductometer COM-360 (HM-digital, Culver City, CA, USA). Bottled water was purchased from the local market.

### 4.2. Standards and Standard Solution Preparation

Individual neat analytical standards of AFL B1 (purity ≥98%), AFL B2, AFL G1, AFL G2, STC (≥98%), T-2 toxin (≥98%), HT-2 toxin (≥98%), DON (≥98%), FB1 (≥98%), FB2 (≥98%), ZEA, (≥98%), and OTA (≥98%) were purchased from Sigma Aldrich (Moscow, Russia); AOH (99.3%), AME (99.77%), ALT (98%), BEA (99.31%), ENN A (99.68%), ENN B (99.62%), MPA (99.59%), and TE (99.84%) were supplied by Fermentek (Jerusalem, Israel). Dry standards were dissolved in an appropriate solvent and diluted to obtain individual stock standard solutions: MPA—2500 μg/mL; DON—1250 μg/mL; ZEA—1000 μg/mL; T-2 and HT-2, AFLs, STE, and OTA—100 μg/mL in acetonitrile; ALT, AOH, AME, TE, BEA, ENN A, and ENN B—200 μg/mL in methanol; FBs—100 μg/mL in acetonitrile–water (50/50%, *v*/*v*). Stock multi-mycotoxin mixed standard solution was prepared in methanol; mycotoxins concentrations were as follows: DON—100 μg/mL; MPA—60 μg/mL; AOH, AME, BEA, ENN A, and ENN B—20 μg/mL; ZEA, ALT—10 μg/mL; T-2 and HT-2—8 μg/mL; FBs—4 μg/mL; TE, STE—2 μg/mL; AFLs and OTA—0.8 μg/mL. It was used for spiking. Dilutions of the stock multi-mycotoxin mixed standard solution in the mix of HPLC mobile phases A and B (1/1 *v*/*v*) were used to obtain external solvent- and matrix-matched calibrations (L1-L7). All stock solutions were stored at −18 °C. The stock multi-mycotoxin standard solution was stored sealed in amber vials for a month. Calibration solutions were prepared fresh before each analysis.

### 4.3. Spiking Procedure and Preparation of Infusions

The effect of infusing period and water characteristics on the transfer rate of mycotoxins was studied for spiked loose green tea (*C. sinensis*) sample. HPLC-MS/MS analysis showed it could be considered as a «clean» matrix for discussed mycotoxins. Tea leaves were used for the preparation of infusions without grinding. Spiking levels were 80 μg/kg for AFLs and OTA; 200 μg/kg for STE and TE; 400 μg/kg for FBs; 800 μg/kg for HT-2 and T-2 toxins; 1000 μg/kg for ZEA, and ALT; 2000 μg/kg for AOH, AME, ENN A, B, and BEA; 6000 μg/kg for MPA; 10,000 μg/kg for DON. These spiking levels correspond to heavily contaminated tea samples reported in the literature (Table 1). High contamination levels afforded direct detection of mycotoxins in the infusion without pre-concentration step and decreased the uncertainty resulting from sample preparation.

The matrix was placed into a glass beaker and spiked with the stock multi-mycotoxin mixed standard solution in the proportion of 100 μL for 1.0 g and left for two hours in darkness to let the solvent evaporate. Water for infusions preparation was heated to 99.9 °C in the thermostat. Indicated below amounts of spiked matrix and heated water were mixed in glass beakers; next, the infusions were cooling without any special treatment. Five grams of the spiked matrix were infused with 250 mL of DW without pH adjustment to study transfer kinetics. Then, 500 μL of the infusion were taken into separate chromatographic vials immediately after pouring hot water and at the 2nd, 5th, 10th, 20th, and 30th minute. Total volume loss was 1.2%. We considered it negligible and did not account for it during transfer calculation. The effect of water TDS and pH on the transfer of mycotoxins was studied by infusing 1.0 g of spiked green tea with 50 mL of water. Experiments were carried out for each kind of water (DW, D-WW, and MW) with pH adjusted to 3–9 with about a unit step (using citric acid and/or sodium carbonate solutions). After pH adjustment, water was heated to 99.9 °C in the thermostat; 50 mL portion was then poured into the beaker with spiked sample and left for 30 min. Then the infusion was paper filtered and analyzed. Each experimental point was repeated in triplicate. Matrix-matched calibrations were used for quantification.

The possible losses in mycotoxins recovery during infusion preparation unrelated to tea matrix, e.g., due to degradation or adsorption on beaker glass, under brewing conditions was evaluated by brewing bare spike. Solvent-matched calibrations were used for quantification.

### 4.4. HPLC-MS/MS

HPLC analysis was performed with a Vanquish UHPLC system consisting of a binary pump, autosampler, and column compartment combined with triple quadrupole mass spectrometer with a heated electrospray source TSQ Endura controlled by Xcalibur 4.0 QF2 Software (all Thermo Scientific, Waltham, CA, USA). The column compartment temperature was 25 °C. The autosampler chamber temperature was set to 5 °C. The mobile phase flow rate was 0.4 mL/min. Injection volume: 4 μL (Titan C18) and 10 μL (Ascentis F5). Mycotoxins were detected under positive ionization mode in the separate runs. Optimized source parameters were as follows: vaporizer temperature 330 °C, positive spray voltage 4500 V, ion transfer tube temperature 225 °C, sheath gas—4 L/min, aux gas—8 L/min; CID gas (argon) 1.5 mTorr; dwell time 100 ms, Q1 and Q3 resolution 1.2 and 0.7 FWHM. Detection was performed in selected reaction monitoring mode (Appendix A).

AFLs, STC, OTA, T-2 and HT-2 toxins, FBs, ZEA, MPA, ALT, AOH, and AME were separated using Ascentis Express F5 (100 × 3.0 mm I.D. pore size 90 Å, particle size 2.7 μm) HPLC column supplied by Supelco (Merck Russia, Moscow, Russia). The mobile phases were constituted of water/methanol (95/5% *v*/*v*, solvent A) and methanol/water (95/5% *v*/*v*, solvent B), both containing 0.1% (*v*/*v*) formic acid and 5 mM ammonium formate. The gradient program was set up as follows: 0–15 min linear gradient from 20 to 95% B; hold at 95% B for 4 min; return to 20% B in 0.3 min; and hold at 20% B for 3.7 min (total run time 23 min). Detection of DON, ENNs, and BEA was much more sensitive with acetonitrile in the mobile phase. The following conditions were set for their detection in the infusions: analytes were separated using Titan C18 (100 × 2.1 mm, pore size 90 Å, 1.9 μm) HPLC column (Merck Russia, Moscow, Russia), under gradient elution. The mobile phases were constituted of: (A) methanol–water (10/90% vol.); (B) methanol–water–acetonitrile (10/10/80% vol.). Both phases were modified with 5 mM ammonium formate and 0.1% formic acid. The gradient scheme was as follows: from the start to 1 min—0% B; from 1 to 2 min—a linear growth to 15% B; from 2 to 5 min—to 30% B; from 5 to 13 min—up to 70% B; from 13 to 14 min—90% B; from 14 to 16.5 min—95% B; up to 17 min—growth to 100% B and then retention for 3 min; from 20 to 20.5 min—a decrease down to 0% B; equilibration—until 22 min.

### 4.5. Method Validation and Data Analysis

Two kinds of calibration curves were used for quantification of mycotoxins in the infusions. Calibration standards prepared “on solvent” were used for bare spike analysis. Mycotoxins in the spiked green tea infusions were quantified using calibration standards prepared “on blank matrix” determination coefficients (R2). The slopes of calibration curves were compared to evaluate matrix effect. For this purpose, the “blank” matrix infusion was prepared in DW without pH adjustment, infusion period 30 min. To evaluate the linearity, seven-point calibration curves were constructed to calculate the signal-to-noise (S/N) approach was used to estimate the LOQ. The chromatographic noise and analytical response were estimated using spiked infusions. The LOQ was defined as S/N = 10. We did not determine the LOD values in the present study, because only quantitative data could be used for transfer estimation. Validation characteristics are presented in the Appendix A.

Average transfer of mycotoxins (except OTA, MPA, and FB2) has been calculated from combined experimental data on their recoveries into infusions prepared using DW, D-WW and NMW with pH adjusted to values from 3 to 9. For the most of mycotoxins 21 points were combined into sets. For AFLs G-type and T-2 toxin we observed decrease of analytes concentration at infusions pH about 7. Literature data suggests their possible degradation under these conditions, which is why corresponding points were excluded from calculation of average transfer. Other outliers were tested by Grubbs criterion. In case analytes could not be quantified in the infusion maximum, the possible transfer rate was evaluated from LOQ.

## Figures and Tables

**Figure 1 toxins-13-00404-f001:**
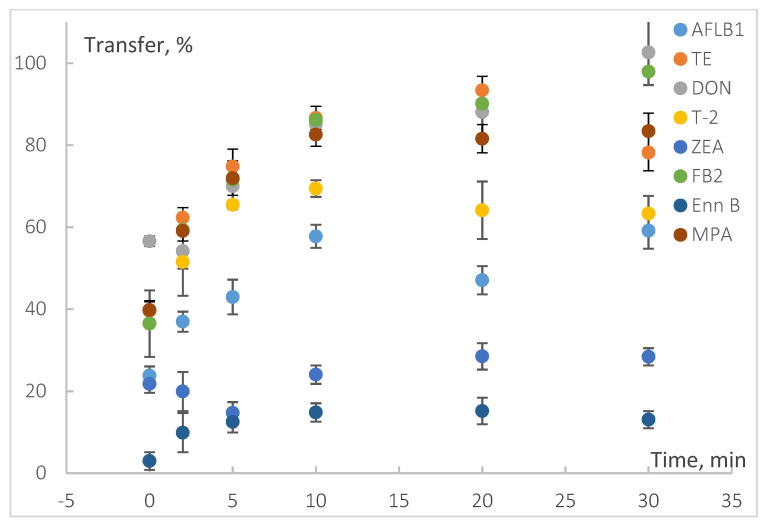
The effect of contact period on the mycotoxins transfers from spiked green tea matrix into infusion.

**Figure 2 toxins-13-00404-f002:**
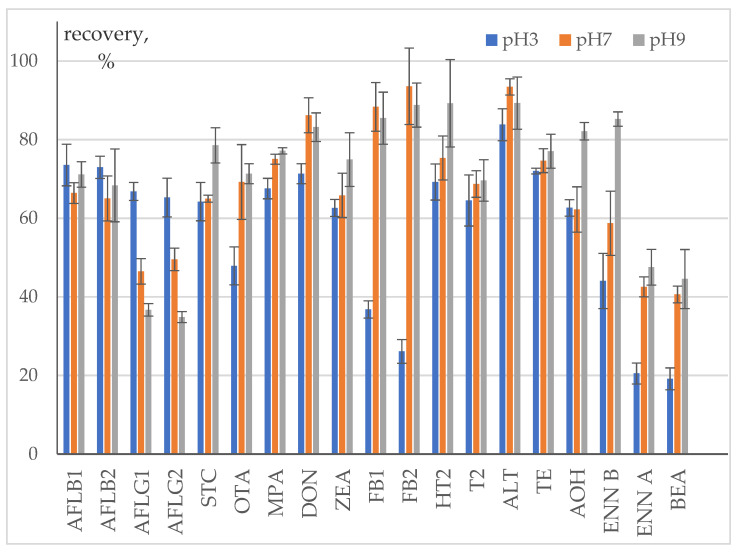
Recovery of mycotoxins from bare spike into an infusion.

**Figure 3 toxins-13-00404-f003:**
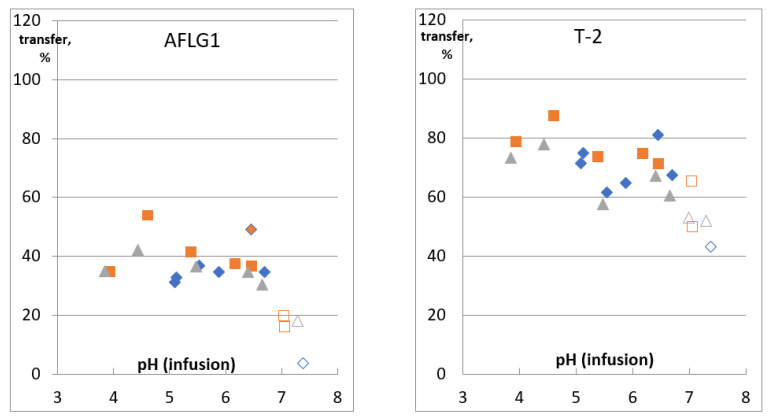
Recovery of AFL G1, T-2, MPA, OTA, and FBs from the spiked matrix into an infusion. 
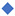
—DW, 
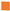
—DW-W, 
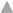
—NMW. Empty symbols indicate points excluded from data set for evaluation of average transfer.

**Figure 4 toxins-13-00404-f004:**
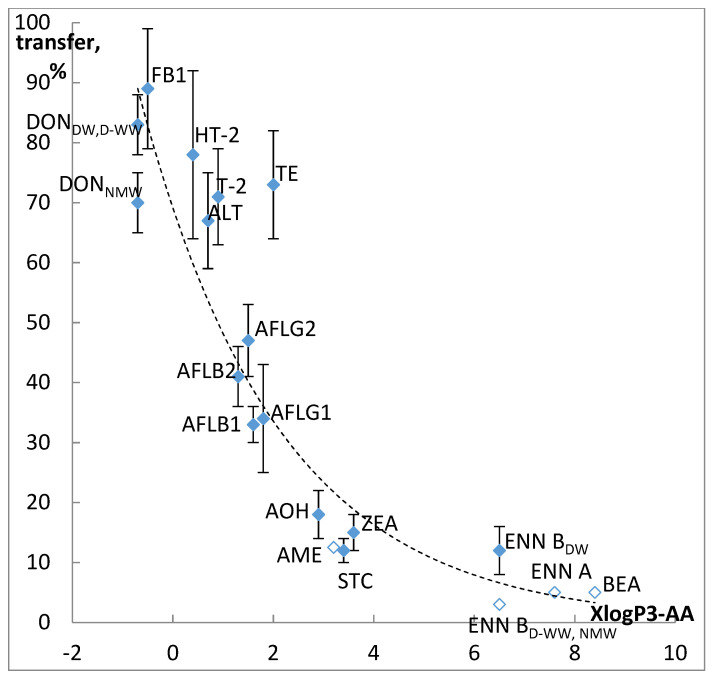
Correlation of mycotoxins polarity and their transfer into the infusion. Filled diamonds—experimental results, empty—maximum transfer predicted from LOQ.

**Table 1 toxins-13-00404-t001:** Occurrence of mycotoxins in *C. sinensis* tea, herbs, herbal teas, and supplements.

Samples, *n* (Origin)	Analytes	Positives			Method	Ref.
		Detected	*n*+, %	Concentration, µg/kg		
*C. sinensis* tea						
Green tea, 129 (Morocco)	AFLs	AFL B1, B2, G1, G2:	59	<0.75–41.8 (B1); 1.2–75.4 (B2), 8–30.2 (G1), 1.3–76 (G2)	HPLC-FLD	[5]
Green tea, 111 (Morocco)	15 mycotoxins: AFLs, OTA, ZEA, *Alternaria* mycotoxins, ENNs, BEA	AOH:	40	max 5.9	HPLC-MS/MS	[6]
		ZEA:	35	max 45.8		
		AFL G1:	2	max 1.6		
		AFL B2:	2	max 7.4		
		ENN B:	2	max 0.3		
		TE:	1	4.6		
Green tea, 15 (Iran)	AFLs, OTA	AFLs:	13	1.0–2.8	HPLC-FLD	[7]
		OTA:	73	0.5–24		
Black tea, 45 (Iran)	AFLs, OTA	AFLs:	40	0.7–4.2		
		OTA:	91	5.1–30.9		
Post-fermented tea, 158 (China)	AFLs	AFL B1:	1 sample	2.1	HPLC-MS/MS	[8]
		AFLs:	1 sample	3.9		
Post-fermented tea, 108 (China)	AFLs, OTA, DON, ZEA, FBs, T-2	OTA:	2	max 36.4	HPLC-UV, HPLC-FLD	[9]
		ZEA:	5	max 182.3		
Pu-erh, 20 (Latvia)	42 myco-toxins	DON and derivatives:	100	max 17,360	2D-LC-TOF	[10]
		AFL B1, B2, G1:	40 (B1)	max 2.7 (B1) and 19.7 (AFLs)		
		OTA/OTB:	25/60	max 4.1/max 30		
		STC:	45	max 93.4		
		ZEA:	35	max 56.1		
		Alternaria mycotoxins	70	- *		
		ENNs	+ *	- *		
Black tea, 63 (Latvia)		DON and derivatives:	39	max 537		
		AFL B1, B2, G1	10	max 3.5 (B1) and 13.9 (AFLs)		
		OTA/OTB:	6/NF	max 7.7		
Green tea, 43 (Latvia)		DON and derivatives:	59	max 3086		
		AFL B1, B2, G1	16	max 3.0 (B1) and 7.7 (AFLs)		
		OTA/OTB:	9/NF	max 3.4/–		
		STC:	7	max 13.0		
		T-2 + HT-2	5	13.7 and 42.4		
		ENNs	+ *	- *		
Green and black tea, 50 (Russia)	29 mycotoxins	MPA:	26	max 200	HPLC-MS/MS	[11]
		STC:	5	<4		
		ENN B:	10	<2.5		
		BEA:	58	<2.5		
Herbs, herbal teas, and supplements						
Herbal tea, 26 (Latvia)	42 mycotoxins	DON and derivatives:	35	max 5631	2D-LC-TOF	[10]
		AFL B1, B2, G1	15	max 15.7 (AFLs)		
		OTA/OTB:	15/15	max 4.2/max 4.6		
		Alternaria mycotoxins,	12	- *		
		ENNs	+ *	- *		
Multicomponent herbal tea, 11 (Russia)	29 mycotoxins	STC	36	8–10	HPLC-MS/MS	[11]
		MPA	45	440–2240		
		TE	54	5.2–9.2		
		ENNs	64	2.8–55		
Chinese herbal medicines (review)		AFL B1	23	0.02–1268.8	--	[12]
		OTA	26	0.01–158.7		
		ZEA	41	0.2–931.07		
		FBs, DON, T-2	+ *	- *		
Chinese traditional medicinal herbs, 48 (China)	AFLs, OTA, CIT	AFLs	71	max 6.3	HPLC-MS/MS HPLC-FLD	[13]
		OTA	29	max 515		
		CIT	8	max 53		
Herbs, 37 (Lebanon)	*Alternaria* mycotoxins (TNZ, TE, ALT, AOH, AME)	TNZ	73	max 4868	HPLC-MS/MS	[14]
		TE	30	max 67		
		AOH	19	max 64		
		AME	54	max 161		
Herbs and herbal tea, 37 (Portugal)	AFLs, ZEA	AFLs	65	2.8–28.2	SPE-ELISA	[15]
		ZEA	62	1.8–19.0		
Herbal dietary supplements, 69 (Czech Republic, US)	57 mycotoxins	DON	23	max: 2890	HPLC-MS/MS	[16]
		T-2/HT-2	38/31	max 1870/max 1530		
		ZEA	52	max 824		
		AOH	69	max 4650		
		AME	78	max 1080		
		TE	46	max 1280		
		TNZ	12	max: 6780		
		ENNs	53–61	max 9260 (ENN B)		
		BEA	64	max 2730		
		MPA	20	max 3260		
		STC	13	max 42		
		OTA	1	max 956		
		PAT	1	max 380		
Medicinal herbs (review)		AFLs	- *	max 380 (B1); max 13.5 (B2);	-	[17]
				max 190 (G1); max 3.2 (G2)		
		OTA		max 253		
		STC		max 20		
		ZEA		max 211		
		FBs		max 237		
		DON		max 344		
		CIT		max 354		
		T-2		max 60.5		

*—is not presented in the original paper and cannot not be calculated from available data; NF—not found; OTB—ochratoxin B; CIT—citrinin; PAT—patulin.

**Table 2 toxins-13-00404-t002:** Mycotoxins’ dissociation constants, experimental (logP) and computed polarity (XlogP3-AA) according to literature data, and average transfer rate into infusion.

Mycotoxin		pKa	logP	XlogP3-AA [37]	Transfer, %
AFL B1	neutral	17.8 [38]	0.45 [38]	1.6	33 ± 3
AFL B2	neutral			1.3	41 ± 5
AFL G1	neutral			1.8	34 ± 9
AFL G2	neutral			1.5	47 ± 6
STC	neutral			3.4	12 ± 2
OTA	amphoteric	3.29/−2.20 [38]	4.4 (pH 3)/1.1 (pH 7) [38]	4.7	38–80 *
MPA	acidic	4.2 [39]		3.2	38–86 *
DON_DW,D-WW_	neutral	11.91 [38]	−1.41 (pH 3)/−1.41 (pH 7) [38]	−0.7	83 ± 5
DON_NMW_					70 ± 5
ZEA	acidic	7.41 [38]	3.83(pH 3)/3.72 (pH 7) [38]	3.6	15 ± 3
FB1	amphoteric	3.64/9.29 [38]	−0.61 (pH 3)/−3.23 (pH 7) [38]	−0.5	89 ± 10
FB2	amphoteric	3.64/9.25 [38]	1.58 (pH 3)/−1.04 (pH 7) [38]	1.2	40–103 *
HT-2	neutral	13.26 [38]	2.27 (pH 3)/2.27 (pH 7) [38]	0.4	78 ± 14
T-2	neutral	13.23 [38]	2.25 (pH 3)/2.25 (pH 7) [38]	0.9	71 ± 8
ALT	acidic	7.5 [40]	0.87 [40]	0.7	67 ± 8
TE	acidic	5.33 [40]	1.21 [40]	2	73 ± 9
AOH	acidic	7.16 [38] 7.63 [40]	3.03 (pH 3)/3.06 (pH 7) [38] 3.18 [40]	2.9	18 ± 4
AME	acidic	6.99 [38] 7.71 [40]	3.93 (pH 3)/3.62 (pH 7) [38] 3.32 [40]	3.2	<12.5 **
ENN B_DW_	basic	−1.08 [38]	3.05 (pH 3)/3.05 (pH 7) [38]	6.5	12 ± 4
ENN B_D-WW,NMW_				6.5	<3 **
ENN A	basic	−0.96 [38]	4.64 (pH 3)/4.64 (pH 7) [38]	7.6	<5 **
BEA	neutral	18.8 [38]	5.5 (pH 3)/5.5 (pH 7) [38]	8.4	<5 **

*—transfer is pH-dependent; **—were not quantified in infusions (maximum transfer evaluated using LOQ).

## Data Availability

The data presented in this study are available in Appendix A.

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
