# Peer review of "Tea: Transfer of Mycotoxins from the Spiked Matrix into an Infusion"

_toxins, 2021, doi:10.3390/toxins13060404_

Round 1
Reviewer 1 Report
Summary and general appreciation:
This study evaluated, by HPLC-MS-MS, the transfer of AFLs, OTA, STE, deoxynivalenol (DON), ZEA, FBs, T-2 and HT-2 toxins, AOH, AME, TE, ENN A and B, beauvericin (BEA) and MPA from the spiked green tea matrix into an infusion under variation of preparation time and water characteristics (total dissolved solids (TDS) and pH).
In my opinion the study and the manuscript as well, were well designed and structured. Moreover, this issue falls in the scope of the journal. My only concern regards the lack of quality control procedures and validation results. Parameters such as linearity, matrix effect, accuracy and precision must be included and discussed in the manuscript or, at least, be included as supplementary material. In the case that the analytical methodology was previously validated in a previous study, a reference must be provided.
In this sense I think the study should be accepted only after the inclusion of the above mentioned items.
Author Response
RESPONSE TO THE COMMENTS AND RECOMMENDATIONS ON THE MANUSCRIPT
Tea: Transfer of Mycotoxins from the Spiked Matrix into an Infusion
The authors would like to thank the reviewers for their attention and expert consideration of the submitted manuscript. We tried to take into account all comments and recommendations, changes were highlighted yellow.
REVIEWER 1
In my opinion the study and the manuscript as well, were well designed and structured. Moreover, this issue falls in the scope of the journal. My only concern regards the lack of quality control procedures and validation results. Parameters such as linearity, matrix effect, accuracy and precision must be included and discussed in the manuscript or, at least, be included as supplementary material. In the case that the analytical methodology was previously validated in a previous study, a reference must be provided.
Data concerning method validation has been added to the manuscript (Lines 436-447) and Supplementary (Table S7). We could not characterize mycotoxins quantification precision as it is usually done within method development and validation procedure. Our aim in this study was to investigate mycotoxins transfer, that is recovery, under variation of infusion preparation conditions. The “clean” matrix used for infusions preparations has been spiked, so concentration of mycotoxins in the matrix is known. Concentration of mycotoxins in the infusions has been evaluated using standard solutions prepared using the infusion of the same matrix prepared under the same conditions. Infusions analysed without any sample preparation.
We have added information concerning linearity range, LOQs, repeatability of HPLC-MS/MS quantification. LODs were not determined as far as we used only quantitative results for the calculation of transfer rate.
Reviewer 2 Report
- In my opinion, the Introduction section should be more concise (consider showing the previously reported results in a form of a table).
- Consider citing this review on a similar topic: Residues and contaminants in tea and tea infusions: a review (DOI: 10.1080/19440049.2014.958575)
- L209-210: The sentence: „AME, ENN A and BEA have not been quantified in the infusions; their transfer rate was evaluated using LOQ“ is rather unclear. I suggest at least changing to „their maximal transfer rate“
- Table S3 is not mentioned in the text.
- Why the pH-related changes of transfer rates are not shown for AFL G1 and AFL G2 (L224)?
- T-2 transfer rate seems to decrease slightly with pH of the infusion. How did you determine the significance of the changes? (adding a paragraph about statistical analysis into the Methods section would be helpful)
- How did you identify outliers? (please mention it in the paragraph about statistical analysis)
- L227: I suggest changing „the increased transfer rate“ to „a higher transfer rate“
- The value 0,86 is R2 (L288) or correlation coefficient (L266)?
- Was the TDS value not changed by using bicarbonate salt to change pH values (L298)?
- I suggest transferring L333-340 in the Discussion part of the text.
Author Response
RESPONSE TO THE COMMENTS AND RECOMMENDATIONS ON THE MANUSCRIPT
Tea: Transfer of Mycotoxins from the Spiked Matrix into an Infusion
The authors would like to thank the reviewers for their attention and expert consideration of the submitted manuscript. We tried to take into account all comments and recommendations, changes were highlighted blue.
REVIEWER 2
In my opinion, the Introduction section should be more concise (consider showing the previously reported results in a form of a table).
Agree. Data on occurrence and maximum levels of contamination is presented in the table in the last version of the manuscript (Table 1, pp. 2-3)
Consider citing this review on a similar topic: Residues and contaminants in tea and tea infusions: a review (DOI: 10.1080/19440049.2014.958575)
There is little information concerning mycotoxins transfer in the above paper. But there are some considerations about factors affecting transfer of contaminants. The review has been cited. Thank you.
L209-210: The sentence: „AME, ENN A and BEA have not been quantified in the infusions; their transfer rate was evaluated using LOQ“ is rather unclear. I suggest at least changing to „their maximal transfer rate“
Corrected.
Table S3 is not mentioned in the text.
Corrected. (Line 252)
Why the pH-related changes of transfer rates are not shown for AFL G1 and AFL G2 (L224)?
We intended to show only several examples, all others are available in the supplementary materials.
However, it may be really more convenient to provide all discussed dependencies in the manuscript body. Figures for AFL G1 and T-2 were added. We decided not to add figure for AFL G2, because it is similar to that of AFL G1.
T-2 transfer rate seems to decrease slightly with pH of the infusion. How did you determine the significance of the changes? (adding a paragraph about statistical analysis into the Methods section would be helpful)
Thank you for this comment. I have overlooked T-2 somehow. Probably, it degrades at pH≥7, thus its concentration decreases. This issue is now addressed to in Lines 270-274.
How did you identify outliers? (please mention it in the paragraph about statistical analysis)
Added. Lines 448-456.
L227: I suggest changing „the increased transfer rate“ to „a higher transfer rate“
Corrected.
The value 0,86 is R2 (L288) or correlation coefficient (L266)?
It is R2. Corrected.
Was the TDS value not changed by using bicarbonate salt to change pH values (L298)?
You are right. It increased. Especially of deionized water. I really have no idea how to increase water pH without changing its TDS and so, that it stays suitable for preparation of tea infusions.
I suggest transferring L333-340 in the Discussion part of the text.
This information has been transferred to Table 1.
Reviewer 3 Report
Tea as the most common daily beverage after pure water is under considerations of mycotoxin contamination. However, the procedure of tea processing and preparing of beverages may have impact on the contamination risks. Despite of several recent studies indicating transfer of AFs, OTA, Fusarium mycotoxins including the emerging and non-regulated metabolites of fungi, and their moderate to high levels in teas, there is need for a proper estimation of kinetics and physicochemical factors affecting the transfer of these toxins from dry mater into tea extracts during preparation of beverages.
It is important topic not only regards the consumer safety , but is also interesting from the perspective of tea extract usage for mycotoxin removal.
The transfer of twenty mycotoxins from the spiked green tea matrix into an infusions
was studied under variation of preparation period, water salinity and infusion pH. The mycotoxins included most of the recently reported mycotoxins as contaminants of dry medical and traditional black and green teas and their infusions.
The manuscript is well organised. The introduction part presents an overview of several reports indicating the rather frequent contamination of teas with multi-class mycotoxins. As discussed, authors of different studies have indicated to different transfer rates and risk assessment procedures. Thus, this study provide good overview and more detailed discussion of the factors affecting transfer rate of mycotoxins into the beverages.
Page 1, lines 16-17 (the discussion of the concentration change are discussed regards the kinetics after the first 10 min period. Is it wright?)
These results are in good coincidence with the several reports of other authors that indicate polarity, contact time as one of the main factors affecting transfer rate.
It has been discussed that there are special traditions of proper preparing green, oolong, Pu-erh and other C. sinensis tea beverages.
In many cases the temperature plays an important role as well as the contact period. Sometimes the first portion of beverage is flushed away, that is important aspect from the removal of mycotoxin extract. These factors are interesting regards the tea quality and safety. It would be recommended to report of these factors or at least to include them in the discussion.
Page 3, lines 122-124: please ad more precisely the purpose of this study. Besides the scope, the gaps solved in this article, should be emphasized.
However, it is nonclear what is the impact of temperature, while we know that most of the mycotoxins are stable at the temperatures of dry tea processing, as well at the temperatures of making infusions, the potential effect of temperature not only the PH on the transfer rate could be mentioned by the authors.
However, the main factors such as infusion-matrix contact period, water TDS and infusion pH as well as the polarity of tea matrix have been well discussed in this paper.
The infusing period is commonly the first factro attributed to different tea making traditions and even culture traditions.
The data for DON indicate more than 100% of DON transfer at 30 min, what is the repeatability of this result?
It is recommended to provide more detailed overview of there results. The supplementary data indicate rather high levels of Alternaria mycotoxins.
As well it is interesting that STC has rather low level compared to AFB1, what is the explanation?
What was the temperature during preparing of the infusions?
Maybe it is worth to evaluate the level of repeated using tea (commonly the first infusion may be flushed away).
Please correct the Fig. 3 and Fig. 4.
Author Response
RESPONSE TO THE COMMENTS AND RECOMMENDATIONS ON THE MANUSCRIPT
Tea: Transfer of Mycotoxins from the Spiked Matrix into an Infusion
The authors would like to thank the reviewers for their attention and expert consideration of the submitted manuscript. We tried to take into account all comments and recommendations, changes were highlighted green.
REVIEWER 3
Page 1, lines 16-17 (the discussion of the concentration change are discussed regards the kinetics after the first 10 min period. Is it wright?)
Yes. That is right.
In many cases the temperature plays an important role as well as the contact period. Sometimes the first portion of beverage is flushed away, that is important aspect from the removal of mycotoxin extract. These factors are interesting regards the tea quality and safety. It would be recommended to report of these factors or at least to include them in the discussion.
Thank you for the comment. This issue has been discussed. Lines 183-187.
Page 3, lines 122-124: please ad more precisely the purpose of this study. Besides the scope, the gaps solved in this article, should be emphasized.
Corrected. Lines 137-142.
However, it is nonclear what is the impact of temperature, while we know that most of the mycotoxins are stable at the temperatures of dry tea processing, as well at the temperatures of making infusions, the potential effect of temperature not only the PH on the transfer rate could be mentioned by the authors.
Thank you. Added. Lines 197-218.
The data for DON indicate more than 100% of DON transfer at 30 min, what is the repeatability of this result?
Almost complete DON transfer has been observed for the infusion of bare spike without matrix. Infusing of the spiked tea resulted in 70-83% recovery (Table 2).
As well it is interesting that STC has rather low level compared to AFB1, what is the explanation?
STC is more hydrophobic in comparison with AFLs: logP 3.4 vs logP 1.3-1.8, that is why it prefers to stay in matrix.
What was the temperature during preparing of the infusions?
Water has been heated to 99.9 C. After pouring hot water into beaker with tea it was left without any special conditions. This detail has been added to the Materials and Methods Line 388-389.
Maybe it is worth to evaluate the level of repeated using tea (commonly the first infusion may be flushed away).
Thank you. It is a good idea for further study. We have mentioned it in discussion Lines 183-187.
Please correct the Fig. 3 and Fig. 4.
Figures captions have been corrected. I hope symbols will not be lost in PDF.
Round 2
Reviewer 1 Report
I think the paper can be published in its present form